# Vertical two-dimensional layered fused aromatic ladder structure

Hyuk-Jun Noh[1], Yoon-Kwang Im[1], Soo-Young Yu[1], Jeong-Min Seo[1], Javeed Mahmood [1✉], Taner Yildirim[2✉] & Jong-Beom Baek [1✉]

Planar two-dimensional (2D) layered materials such as graphene, metal-organic frameworks, and covalent-organic frameworks are attracting enormous interest in the scientific community because of their unique properties and potential applications. One common feature of these materials is that their building blocks (monomers) are flat and lie in planar 2D structures, with interlayer π–π stacking, parallel to the stacking direction. Due to layer-to-layer confinement, their segmental motion is very restricted, which affects their sorption/desorption kinetics when used as sorbent materials. Here, to minimize this confinement, a vertical 2D layered material was designed and synthesized, with a robust fused aromatic ladder (FAL) structure. Because of its unique structural nature, the vertical 2D layered FAL structure has excellent gas uptake performance under both low and high pressures, and also a high iodine ($I_2$) uptake capacity with unusually fast kinetics, the fastest among reported porous organic materials to date.

[1] School of Energy and Chemical Engineering, Center for Dimension-Controllable Organic Frameworks, Ulsan National Institute of Science and Technology (UNIST), 50 UNIST, Ulsan 44919, South Korea. [2] Center for Neutron Research, National Institute of Standards and Technology, Gaithersburg, MD 20899, USA. ✉email: javeed@unist.ac.kr; taner@nist.gov; jbbaek@unist.ac.kr

The discovery of graphene in 2004[1] and its interesting intrinsic characteristics[2] has stimulated the design and synthesis of two-dimensional (2D) layered structures with unique properties[3]. Around the world, such research has largely focused on the synthesis of planar 2D layered metal-organic frameworks (MOFs), covalent organic frameworks (COFs) and porous organic networks (PONs). Planar 2D layered MOFs, which contain metal cores and organic linkers, show outstanding sorption capacity with high crystallinity, but their poor stability in moisture impedes practical applications[4]. Planar 2D layered COFs and PONs have also been designed and synthesized using various combinations of building blocks (monomers)[5]. Their frameworks are primarily constructed of covalent bonds between building blocks, however, most reported planar 2D layered COFs and PONs also contain weak, reversible –B–O– and –N=C– bonds, which have poor stability against hydrolysis, limiting their practical applications. To overcome the problems associated with unstable MOFs, COFs, and PONs, recent efforts have focused on synthesizing stable 2D organic network structures[6].

Such stable 2D organic networks have a number of beneficial features, including custom designability, light weight, permanent porosity and high thermal and chemical stability. Their meticulously designed structures are promising for practical applications in catalysis[7], energy conversion and storage[8], electronics[9], gas storage[10] and separation[11] and the uptake of dangerous chemicals[12]. In particular, robust 2D fused aromatic network (FAN) structure composed of fully π-conjugated linkages with evenly distributed holes and nitrogen atoms ($C_2N$) has displayed outstanding performance for application in electronics[9]. In addition, other FAN structures cocooning Fe nanoparticles have shown high potential as electrocatalysts[13,14].

To date, however, reported 2D COFs[15], PONs[16,17], and FANs[9] still have mostly planar structures, with building block units which are flat, and lie parallel to the layer stacking direction, to optimize interlayer π–π stacking. These planar layered stacking patterns restrict segmental motion in the structures, leading to poor sorption/desorption kinetics when used as sorbent materials. And depending on the particular stacking pattern, available pores can also be blocked by adjacent layers, which further reduces available active surface area and pore volume.

A more effective approach would be to design vertical 2D layered structures, where the structural repeating units are oriented perpendicularly to the layer stacking direction. With this configuration, it would be possible to minimize interlayer contact while maximizing segmental motion. This would lead to high available surface area, and fast sorption/desorption kinetics.

Such vertical 2D layered structures have been realized using robust poly(benzimidazobenzophenanthroline), or BBL polymer. Since the first linear BBL polymer[18], many other linear BBL structures have been synthesized using different preparation methods[19] and have exhibited various physical and chemical properties[20]. They have been developed for lithium-ion battery applications as an anode material[21], for organic semiconductors[22] and transistors[23].

Recently, our group reported unique planar 2D layered BBL structures[24] which showed good performance for the removal of toxic ions. In the present work, we report the synthesis of a vertical 2D layered BBL (designated V2D-BBL) structure, and its unique sorption behaviors. A unique nature of V2D-BBL is vertically standing structural units, which attribute to minimizing interlayer contact points and maximizing available exposed surface area. As a result, the V2D-BBL exhibits an excellent gas uptake performance under both low and high pressures, and a high iodine ($I_2$) uptake capacity with the fastest kinetics among reported organic porous materials to date. Hence, this present work can be exploited as an unique class of 2D structure.

## Results

**Synthesis and characterization of vertical 2D layered BBL.** The V2D-BBL structure was prepared from a simple polycondensation between triptycene hexamine (THA) and naphthalenetetracarboxylic dianhydride (NDA) in polyphosphoric acid (PPA) (Fig. 1a). The detailed synthesis process is described in the Method section and illustrated in Supplementary Figure 1. The driving force for the structure-forming reaction is aromatization with a huge thermodynamic energy gain. The resultant BBL structure has the properties of a tailored extended π-conjugated aromatic skeleton with periodic heteroatoms (N and O) in each structural unit. The distinct vertical 2D layered structure is associated with the triptycene unit in the THA (Fig. 1a). The THA unit provides a vertically laid-up structure with minimum contact between layers, leading to maximum exposure of the available active surface area[25] and maximum freedom for segmental motion, resulting in fast reversible sorption/desorption kinetics[26].

In conventional planar 2D layered structural units (Fig. 1b), molecular stacking is promoted by strong interlayer π-π interactions, which consequently block off available surface area. In contrast, the vertical 2D layered structural units (Fig. 1c) minimize interlayer contact points, and thus maximize internal free volume and the available active surface area. Although both structures have the same numbers of units (Fig. 1b, c), the vertical 2D layered structure is expected to have much higher surface area, volume and segmental flexibility.

To confirm the detailed structure, the formation of benzimidazole rings and pyrrolidinone moieties were investigated using Fourier-transform infrared (FT-IR) and solid-state $^{13}C$ cross-polarization magic-angle spinning (MAS) nuclear magnetic resonance (NMR) spectroscopy. To assertain the conversion from a rotatable intermediate imide form (before complete cyclization) to a closed aromatic BBL ring (after complete cyclization), the porous imide network (PIN) precursor was collected by quenching the reaction at 150 °C (Supplementary Fig. 1c). At that point, the PIN structure, composed of a single bonded network (rotatable), has a short conjugation length and a deep-red color (Fig. 1a, Supplementary Fig. 1c). After complete cyclization, the reaction mixture becomes dark-purple (Supplementary Fig. 1d), indicating the expansion of conjugation length.

The appearance of the C=N stretching band at 1628 cm$^{-1}$ supports the formation of the benzimidazole ring in the BBL structure (Fig. 2a)[27,28]. Moreover, the carbonyl (C=O) stretching vibration peak located next to nitrogen shifted from 1706 to 1698 cm$^{-1}$ after cyclization, because delocalization of the π-electrons from the extended aromatic structures of the fused ring moiety reduces the electron density of the carbonyl double bonds. With these peak changes, the other peaks are well overlapped, containing aromatic C=C stretching (1552 cm$^{-1}$), C–N–C moiety (1384 cm$^{-1}$) and unreacted carbonyl moiety (1780 cm$^{-1}$) at the edges[29].

In addition, the formation of a fused aromatic BBL ring was further confirmed using solid-state $^{13}C$ CP-MAS NMR spectra (Fig. 2b). Both samples have three similar broad peaks with chemical shifts of 114 (c, e and c', e'), 128 (b, g and b', g') and 160 ppm (f and f'), which can be assigned to the aromatic $sp^2$ carbons and carbonyl (C=O) groups in their structure[24,27]. The clear differences between them are that the PIN displays a carbon peak of N–C moiety in the imide bond at 142.6 ppm (d) and an $sp^3$ bridge carbon peak at 53.5 ppm (a). On the other hand, the V2D-BBL structure shows a carbon peak of N−C moiety in the benzimidazole ring (d') at 144.1 ppm and the $sp^3$ bridge carbon peak (a') at 56.2 ppm. The up-field shift of the V2D-BBL structure, which suggests more deshielding, can be attributed to the inductive effect from the formation of fused aromatic rings and two eletronegative nitrogen atoms in each benzimidazole

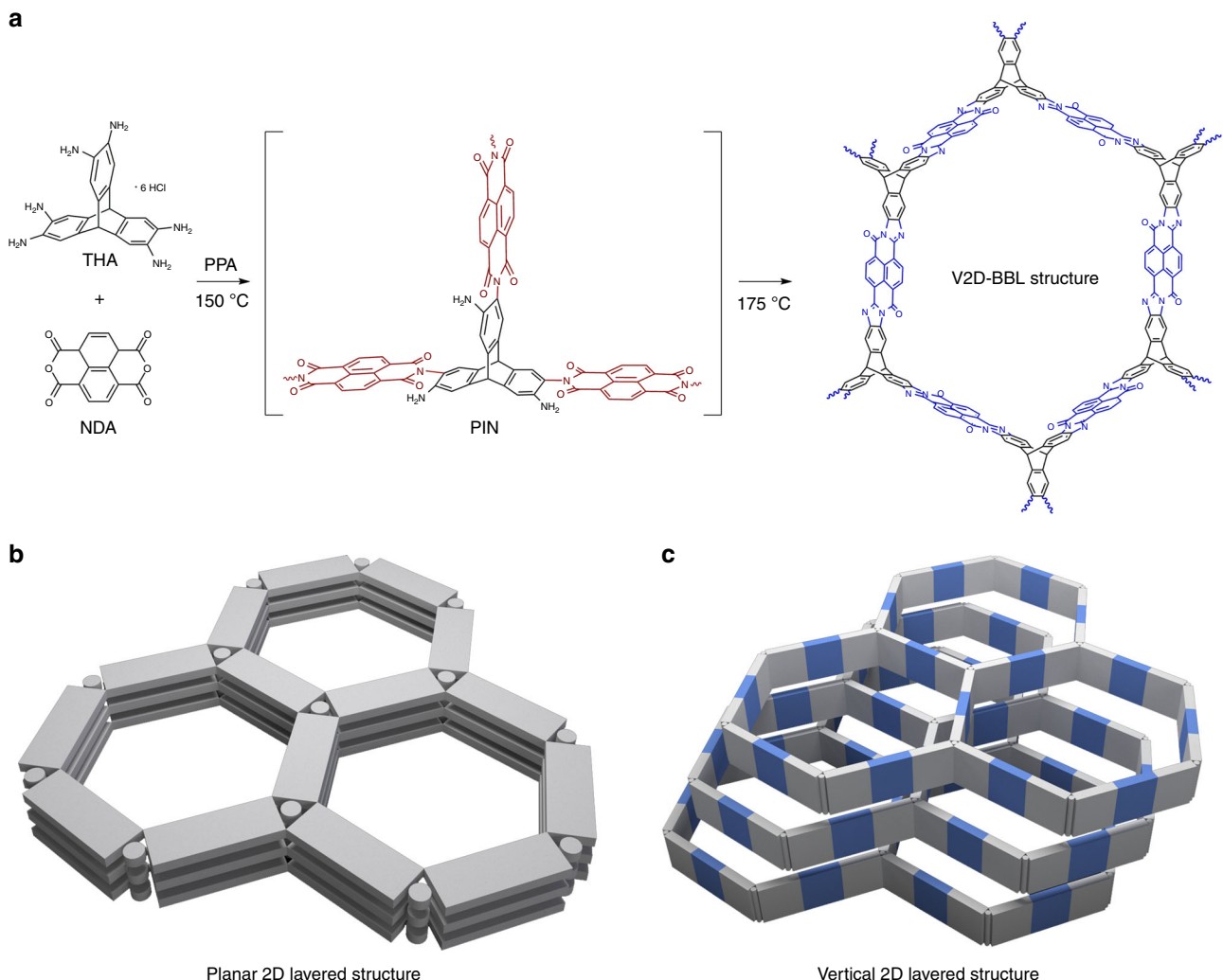

**Fig. 1 Synthesis of the vertical 2D layered structure. a** Schematic illustration of the formation of the vertical 2D layered BBL structure, from the reaction between THA and NDA as building blocks. Representative structural comparison of the planar and vertical 2D layered structures: **b** stacking pattern of the conventional planar 2D layered structural units and **c** stacking pattern of the vertical 2D layered structural units.

ring[30,31]. On the basis of these results and the visual observation of transitions during the reaction (Supplementary Fig. 1), the fomation of the designed V2D-BBL structure could be confirmed.

X-ray photoelectron spectroscopy (XPS) was used to look into the bonding nature of the V2D-BBL structure. The XPS survey spectra unfolded only three peaks (Fig. 2c) related to carbon (C 1 s), nitrogen (N 1 s) and oxygen (O 1 s), without any other elements. Deconvolution of the N 1 s resulted in two peaks, associated with the tertiary N and C=N–C at 400.6 and 398.5 eV, repectively[32]. This supports the formation of the benzimidazole rings again (Inset of Fig. 2c). The deconvoluted C 1 s and O 1 s spectra are also in agreement with the expected structure (Supplementary Fig. 3, Supplementary Note 1).

The thermal stability of the samples was evaluated by thermogravimetric analysis (TGA) both under air and nitrogen atmospheres (Fig. 2d). The thermal stability of the V2D-BBL structure was better than that of the PIN precursor (Supplementary Fig. 4). The structure exhibited superior thermoxidative stability with a weight loss of only 4%, from 50 to 500 °C, even under air atmosphere. This high thermal stability suggests that an extremely high molecular weight (ca. ∞) compound was formed with minimal edge effect. In addition, the decomposition temperature above 600 °C under nitrogen atmosphere is also

associated with the formation of the stable fused aromatic BBL structure[33].

Field-emission scanning electron microscopy (FE-SEM) was used to examine morphology of the V2D-BBL structure, which shows clean stacked sheet-like texture and few tens of micrometer grain size (Supplementary Fig. 6a). Moreover, the high-resolution transmission electron microscopy (HR-TEM) images also display clean sheets with layered structure and uniform micropores throughout the sample (Supplementary Fig. 6b).

The quantitative elemental composition of the sample was also confirmed by elemental analysis (EA), energy dispersive X-ray spectroscopy (EDS) coupled with SEM (SEM-EDS) and EDS elemental mappings, and XPS. Elemental mapping showed that carbon, nitrogen and oxygen elements were well-distributed in the structure (Supplementary Fig. 5). The overall elemental composition of the sample described by different techniques is summarized in Supplementary Table 1, showing good agreement between theoretical and experimental values.

The bulk crystallinity of the sample was evaluated using powder X-ray diffraction (PXRD) patterns. The V2D-BBL structure showed broad PXRD peaks (Supplementary Fig. 7), which implies a low ordering nature, the severe vibration of vertically aligned segments, or infinite molecular weight (ca. ∞)

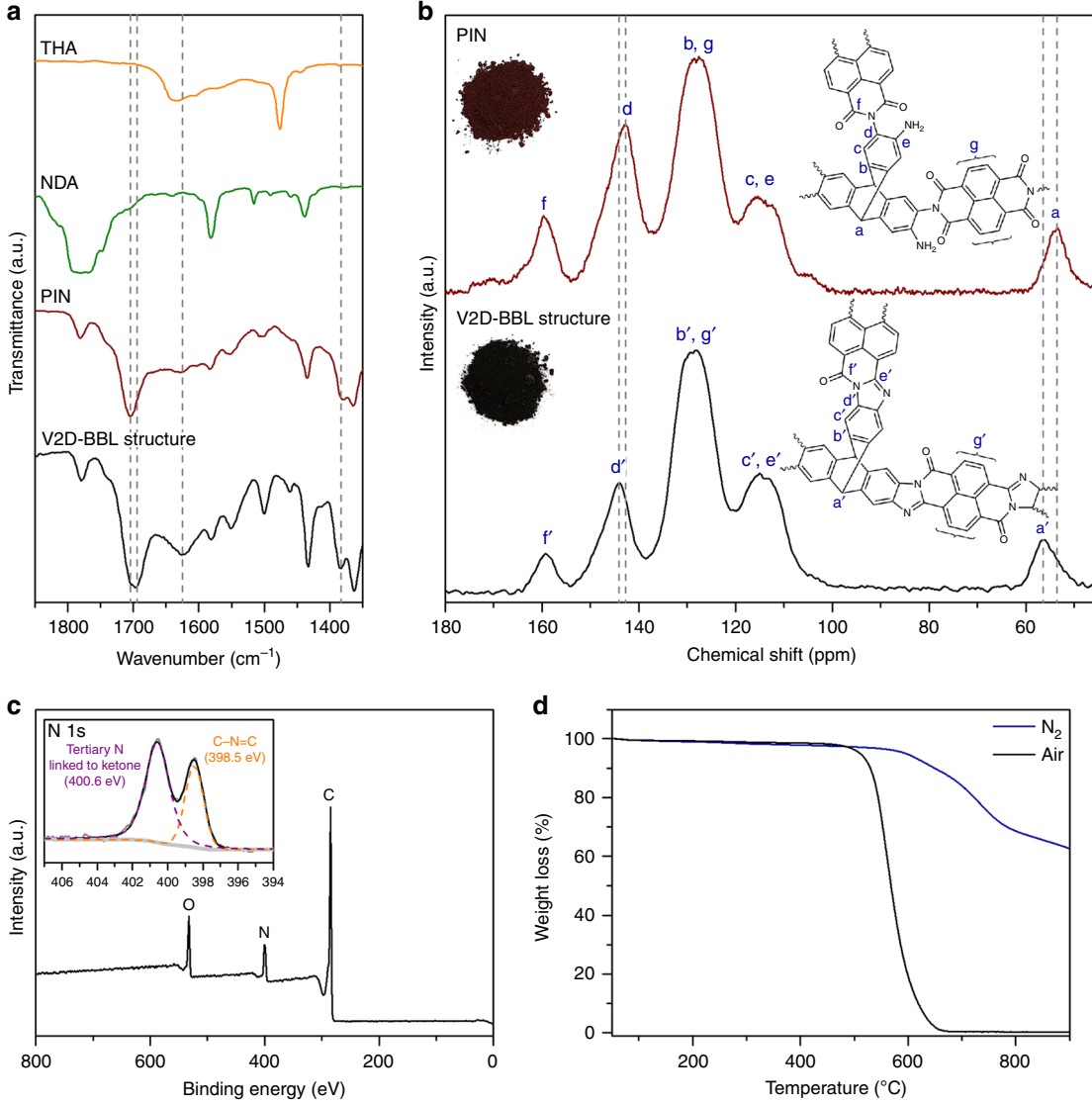

**Fig. 2 Characterizations and thermal property of the V2D-BBL structure. a** Fourier-transform infrared (FT-IR) spectra (KBr pellets) of THA (orange line), NDA (green line), intermediate PIN (wine line) and V2D-BBL structure (black line). Spectra were magnified in the range of 1350–1850 cm$^{-1}$, highlighting the formation of benzimidazole rings (Full spectra and corresponding assignments are shown in Supplementary Fig. 2. **b**, Solid-state $^{13}$C CP-MAS NMR spectra: Top-right inset: the structure of an intermediate PIN sample. Top-left inset: a photograph of a powder PIN sample. Bottom-right inset: the structure of the V2D-BBL sample. Bottom-left inset: a photograph of the V2D-BBL sample. **c** XPS survey of the V2D-BBL sample. Inset: deconvoluted N 1s spectrum. **d** Thermogravimetric analysis (TGA) curves of the V2D-BBL sample under air (black) and N$_2$ (navy) atmospheres at a ramping rate of 10 °C m$^{-1}$.

driven by the powerful aromatization, or all of these features together. This could be the result of poor interlayer interactions between vertical layers during the solution synthesis, because the vertical layered structures have very weak contact points. On the other hand, conventional planar 2D layered structures have strong layer to layer interactions, which produce comparatively better crystallinity. Still, the local ordering with broad peaks, which is quite different from the interlayer interactions of planar 2D layered materials such as graphite and *h*-BN. It suggests that the peak broadening may be, in significant part, associated with a randomly oriented stacking nature, and scattering induced by the vibration of structural units, which are vertically laid-up in the BBL structure and thus have a relatively high degree of freedom.

**Gas sorption performances under both low and high pressure.** To examine the permanent porosity and surface area of the samples, the nitrogen isotherm was measured via the Brunauer–Emmett–Teller (BET) method at 77 K. As shown in

Fig. 3a, the nitrogen (N$_2$) adsorption-desorption isotherm displays steep adsorption in the low-pressure range (0− 0.01), indicating the microporous nature of the material according to the IUPAC classification[34]. The large hysteresis between adsorption and desorption is associated with narrow pores, contributing to the capillary effect. The BET specific surface area ($S_{BET}$) was calculated to be 1724 m$^2$ g$^{-1}$ and the total pore volume was 1.37 cm$^3$ g$^{-1}$ (Supplementary Fig. 8). Compared to its planar 2D layered analogues ($S_{BET}$ = 365–615 m$^2$ g$^{-1}$)[24], the high $S_{BET}$ value of V2D-BBL is attributed to its rigid vertical nature, which increases the accessible surface area with open pores, by minimizing interlayer π-π interactions.

The surface area and pore volume of the V2D-BBL structure were found to be comparable to many reported porous 2D and 3D organic materials (Supplementary Table 2). The non-linear density functional theory (NLDFT) pore size distribution plot (inset of Fig. 3a) reveals that a sharp micropore peak was centered at 0.56 nm with a minor broad peak at around 0.87 nm.

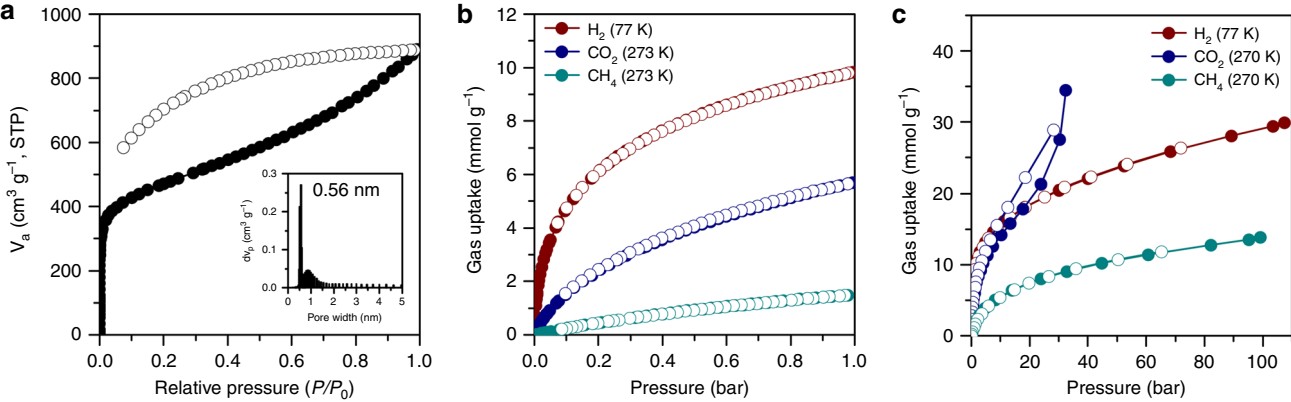

**Fig. 3 BET data and gas sorption study of the V2D-BBL layered structure. a** Nitrogen adsorption-desorption isotherm obtained at 77 K and 1 bar. Inset: corresponding non-linear density functional theory (NLDFT) pore size distribution. **b** Low-pressure and **c**, high-pressure gas ($H_2$, $CO_2$, and $CH_4$) sorption isotherms of the V2D-BBL structure at different temperatures.

Recently, the development of efficient and stable gas ($H_2$, $CO_2$ and $CH_4$) storage methods has become increasingly important, as promising alternatives to conventional fossil energy, because of the relative clean, sustainable and economical benefits[35,36]. Since the newly formulated samples exhibited remarkable porosity and an attractive structure, a gas adsorption study was conducted in low-pressure conditions (Fig. 3b, Supplementary Fig. 9).

At low-pressure, the V2D-BBL structure showed significant gas uptake, with values of 9.84 mmol g$^{-1}$ (1.97 wt%) of $H_2$ uptake at 77 K, 5.70 mmol g$^{-1}$ (25.1 wt%) of $CO_2$ uptake and 1.49 mmol g$^{-1}$ (2.39 wt%) of $CH_4$ uptake at 273 K (the detailed values at different conditions are described in Supplementary Note 2). All of the isotherm graphs indicated completely reversible physisorption without hysteresis. Furthermore, all of the uptake amounts were much higher than those reported for planar 2D layered BBL structures in the literatures[24], and rotatable PIN (Supplementary Figs. 10–13, the detailed values are summarized in Supplementary Table 2), and were comparable to other reported porous organic materials (POMs) (Supplementary Table 2). It appeared that the rigid vertical 2D layered BBL structure provides good pore accessibility, which facilitates interaction with gas molecules. As further evidence, all of the isosteric heats of adsorption ($Q_{st}$) for $H_2$, $CO_2$ and $CH_4$ were determined to be high values (Supplementary Fig. 14), indicating good affinities toward gas molecules[37] (see Supplementary Note 3 for more details).

To gain further insight into the material's structural advantages for gas sorption, high-pressure gas adsorption-desorption measurements were also performed, since no saturation in adsorption was observed at 1.0 bar. The $H_2$ isotherms were measured at 77, 160, and 296 K up to 110 bar (Supplementary Fig. 15). The maximum total and excess $H_2$ uptake values measured at 77 K were found to be 29.92 mmol g$^{-1}$ (6.03 wt%, Fig. 3c, Supplementary Fig. 15a) and 13.54 mmol g$^{-1}$ (3.14 wt%, Supplementary Fig. 15b), respectively, with completely reversible isotherms. These values are outstanding among POMs reported in the literature (Supplementary Table 2).

To assess its applicability for practical use as a material for $H_2$ storage, $H_2$ uptake capacity was investigated at 296 K, near room temperature (Supplementary Figure 15a). Interestingly, the extraordinary total $H_2$ uptake of 1.27 wt% was observed. This value exceeds those of some famous reported porous materials[38].

The total $CO_2$ storage capacity was exceptional, reaching 34.5 mmol g$^{-1}$ (1517.5 mg g$^{-1}$, Fig. 3c, Supplementary Fig. 16a) at 270 K and 33 bar. This value seems to be the highest among reported POMs, as summarized in Supplementary Table 2.

The V2D-BBL structure is composed of a nitrogen and oxygen doped fully fused aromatic ring system, which contributes to hysteresis in the adsorption and desorption curves after ~ 5 bar[39]. This value is also a remarkable performance compared to other POMs (Supplementary Table 2). The slight hysteresis could be due to minor chemisorption.

Along with good $H_2$ and $CO_2$ uptake performances, unusual total and excess $CH_4$ uptake values at 270 K and 100 bar were observed, 13.9 mmol g$^{-1}$ (222.5 mg g$^{-1}$, Fig. 3c, Supplementary Fig. 17a) and 8.18 mmol g$^{-1}$ (131.2 mg g$^{-1}$, Supplementary Fig. 17b), respectively. These superior uptake performances are due to the inherently abundant accessible micropores in the V2D-BBL structure. Again, the uptake values were superior to those of most reported POMs (Supplementary Table 2).

**Vapor iodine capture performance.** One of the pressing global safety and environmental issues today is the volume of waste products that are generated during nuclear fission. Because of their structural diversity and rich porosity, POMs have been considered promising candidates for vapor iodine ($I_2$) capture. Radioactive isotopes of iodine, such as $^{129}I$ and $^{131}I$, are a major vapor waste in fission, and $^{129}I$ has a very long radioactive half-life ($1.57 \times 10^7$ years), with permanent environmental implications[12,40]. Efficient methods that can quickly capture radioactive iodine are of great importance to ensure the safe utilization of nuclear energy in the future.

To study iodine vapor adsorption, the V2D-BBL structure was exposed to iodine vapor at 348 K under ambient pressure, which is near to the typical nuclear fuel reprocessing condition. To calculate the amount of iodine adsorbed, the $I_2$@V2D-BBL structure complex was heated to 333 K for 2 h to remove weakly adsorbed iodine on its surface. As indicated in Fig. 4a, the amount of iodine uptake increased rapidly up to 3.0 g g$^{-1}$ within 2.5 h and then saturated. Meanwhile, the rotatable PIN precursor exhibited a lower amount of iodine uptake (1.8 g g$^{-1}$) with no saturation up to 24 h, indicating again the advantage of the V2D-BBL structure. Although the uptake capacity of the V2D-BBL structure was not the highest among the materials reported in the literature (Supplementary Table 3), it did have the fastest saturation time, which is the biggest advantage (Fig. 4b). The reason for the rapid saturation time (the fastest kinetics) can be explained as resulting from the strong interaction between iodine vapor and the nitrogen rich rigid fused aromatic structure[41], and the well-exposed surface with open pores, thanks to the unusual V2D-BBL structure.

Furthermore, distinct iodine capture from solution was observed (see Supplementary Note 4 for more details). The purple colored

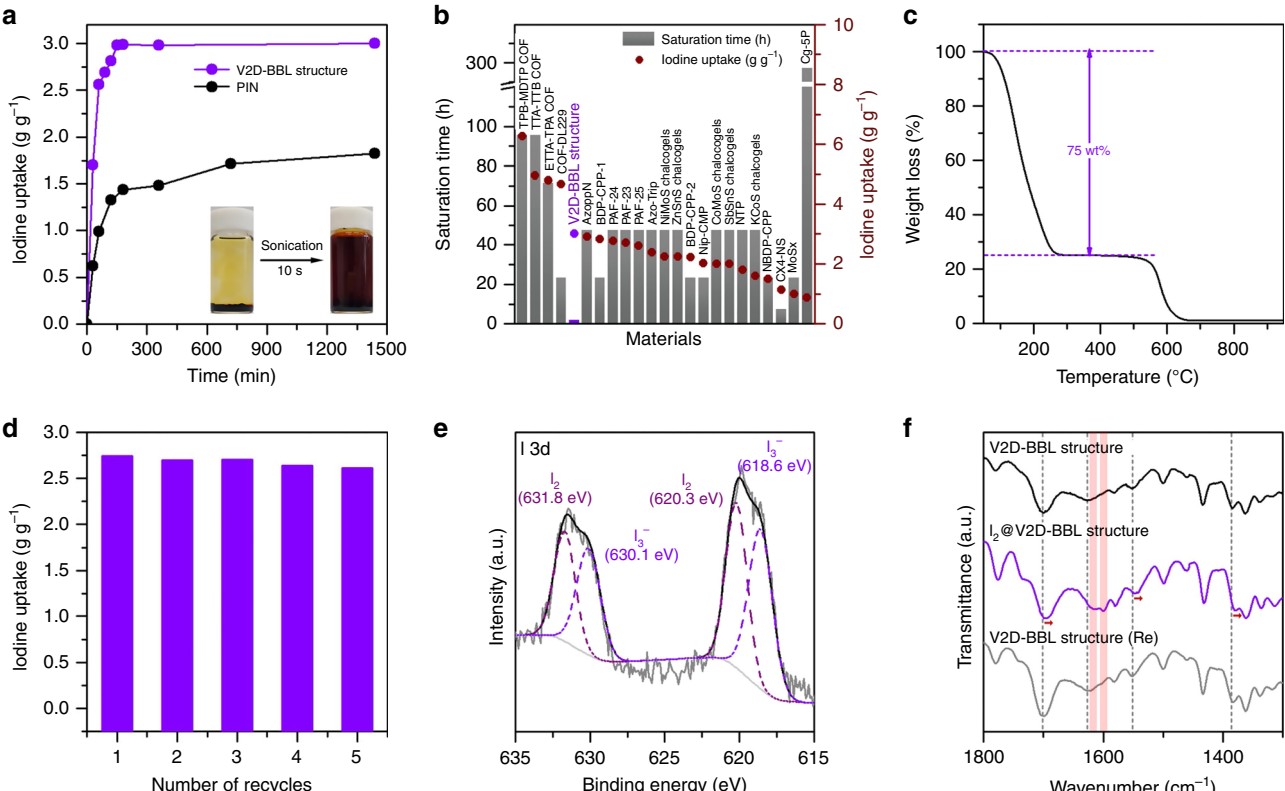

**Fig. 4 Vapor iodine sorption study of the V2D-BBL structure. a** Gravimetric iodine uptake of the V2D-BBL structure (purple) and PIN (black) as a function of exposure time at 348 K and ambient pressure. Inset: the photographs of iodine release in EtOH by sonication for 10 s. **b** Comparison of saturation time and iodine uptake (data taken from Supplementary Table 3). **c** TGA curve of the $I_2$@V2D-BBL structure under air atmosphere with a ramping rate of 10 °C min$^{-1}$. **d** Recyclability of the V2D-BBL structure for iodine adsorption by vapor sublimation at 348 K. **e** Deconvoluted I 3d XPS spectrum. **f** FT-IR spectra of the V2D-BBL structure, $I_2$@V2D-BBL structure and regenerated V2D-BBL structure after five recycles.

solution gradually disappeared and became colorless after 48 h (Supplementary Fig. 18). Thus, the V2D-BBL structure exhibited excellent iodine uptake performance from both vapor and solution.

TGA was used to determine the thermal retention behavior of the $I_2$@V2D-BBL structure under air atmosphere. As shown in Fig. 4c, there were two weight loss steps, related to the release of iodine in the range of 90–270 °C and decomposition of the BBL moiety above 550 °C, respectively, which was the same temperature as the pristine V2D-BBL structure (Fig. 2d). Based on the TGA curve, the weight loss was approximately 75 wt%, which is associated with the evaporation of physically adsorbed iodine. The calculated iodine uptake was 3.0 g g$^{-1}$, which is in good agreement with the maximum uptake value calculated from the weighing balance.

The V2D-BBL structure was regenerated by immersing the $I_2$@V2D-BBL structure complex in ethanol and sonicating for 10 s (Inset of Fig. 4a). After sonication and filtration, no color was observed from the regenerated sample (Supplementary Video 1), indicating facile desorption due to the physisorption dominant system. Then, this regenerated sample was tested for recyclability. Only a small uptake capacity loss was observed even after five repetitions of the recycle test (Fig. 4d). The facile regeneration and high capacity retention after recycling of the V2D-BBL structure were attributed to the nitrogen/oxygen rich fully π-conjugated fused aromatic ring structure, which enabled dominant physisorption with minor chemisorption, as indicated by the high-resolution I 3D XPS spectrum (Fig. 4e, Full XPS survey spectrum is presented in Supplementary Fig. 19).

To examine the affinity of iodine toward the V2D-BBL structure, FT-IR spectra were measured before/after iodine

capture and 5 regenerations. The peak at 1628 cm$^{-1}$ assigned to the C=N bonds in the benzimidazole ring disappeared after iodine adsorption, and new vibration bands appeared at 1615 and 1600 cm$^{-1}$ (Fig. 4f), suggesting that the benzimidazole ring has an affinity toward iodine. Furthermore, peak shifts from 1698 to 1694 cm$^{-1}$ occurred for carbonyl (C=O), C=C bonds in aromatic rings from 1552 to 1546 cm$^{-1}$, C−N−C moiety in benzimidazole ring from 1384 to 1379 cm$^{-1}$. All of the peak shifts were associated with a decrease in electron cloud density, caused by the interaction with iodine[42].

These newly appeared and shifted peaks confirmed the presence of iodine affinity sites, mainly nitrogen atoms in the benzimidazole rings, and carbonyl groups and conjugated aromatic rings. Moreover, the regenerated sample exhibited the same FT-IR spectrum as the pristine sample even after five times of reuse. The result suggests that the V2D-BBL structure has a highly reversible nature for the physisorption of iodine. SEM images and EDS mappings were used to compare the $I_2$@V2D-BBL structure with the regenerated sample, and the results indicated stable and reversible iodine vapor capture, since there was no notable change in morphology and the almost complete removal of iodine was observed (Supplementary Figs. 20–21, Supplementary Note 5).

**Discussion**

In summary, we have developed an attractive type of structure, with vertically aligned fully aromatic repeating units on its edges, with enhanced sorption capacity and high reversibility. The vertical 2D poly(benzimidazobenzophenanthroline) structure (V2D-BBL structure) was realized via a simple wet chemical reaction in gram-

scale. The prepared V2D-BBL structure exhibited outstanding physiochemical and thermal stability, and rigidity due to the fully fused aromatic ring system. Furthermore, its unique vertical 2D layered structure restricts interlayer π-π interaction, enabling maximum surface exposure while providing segmental freedom.

The V2D-BBL structure with uniform heteroatoms (N or O) and robust fused aromatic rings displayed excellent gas uptake performance under both low and high pressures, as well as high iodine uptake capacity ($3.0 \, g \, g^{-1}$) and the fastest saturation time (2.5 h), and reusability (87.2% after five times recycles). Considering its intrinsic high thermal and physiochemical stability, and exceptional sorption capacity, this unique V2D-BBL structure can be a promising platform for high performance sorbent materials for practical applications. The development of the V2D-BBL structure underscores the huge potential for the design and synthesis of more unique structures in future.

## Methods

**Materials**. All the solvents, chemicals and reagents were purchased from Aldrich Chemical Inc., unless otherwise stated. Solvents were degassed with nitrogen purging before use. All reactions were conducted under nitrogen atmosphere with oven dried glassware. Detailed synthesis procedure for hexamine (THA) hexahydrochloride is described in Supplementary Method[39,43].

**Synthesis of the V2D-BBL structure**. Triptycene hexamine (THA) hexahydrochloride (1.0 g, 1.775 mmol) was taken in polyphosphoric acid (PPA) in a special PPA reactor equipped with a mechanical stirrer under nitrogen flow. HCl gas was removed at room temperature for two days and at 50 °C for one day, then, at 70 °C overnight to give a completely transparent solution, free from HCl gas. The removal of the HCl gas was monitored by pH paper at the nitrogen exhaust point. The reaction mixture was cooled to room temperature and 1,4,5,6-naphthalene-tetracarboxylic dianhydride (NDA) (0.714 g, 2.66 mmol) was added and stirred at room temperature for 2 h to ensure complete mixing of the monomers. Then the temperature was raised to 60, 100, and 150 °C for 2 h each. Finally, the temperature was increased to 175 °C. The structure formation was complete within 2 h. The reaction was stopped after complete solid-gel formation. The reaction mixture was cooled down to room temperature and precipitated in water to remove PPA from the material. After filtration the solid product was further Soxhlet extracted with water and methanol for 3 days each to completely wash off impurities, if any. Then the product was freeze dried at –120 °C for 3 days, resulting in dark-purple powder (1.35 g, 99.1% isolated yield).

**Synthesis of the PIN**. Triptycene hexamine (THA) hexahydrochloride (1.0 g, 1.775 mmol) was taken in polyphosphoric acid (PPA) in special PPA reactor equipped with a mechanical stirrer under nitrogen flow. HCl gas was removed at room temperature for two days and 50 °C for one day, then at 70 °C overnight to give a completely transparent solution, free from HCl gas. The removal of the HCl gas was monitored by pH paper at the nitrogen exhaust point. The reaction mixture was cooled to room temperature and 1,4,5,6-naphthalenetetracarboxylic dianhydride (NDA) (0.714 g, 2.66 mmol) was added and stirred at room temperature for 2 h to ensure complete mixing of the monomers. Then the temperature was raised to 60, 100, and 150 °C for 2 h each. The reaction mixture was cooled down to room temperature and precipitated in water to remove PPA from the material. After filtration the solid product was further Soxhlet extracted with water and methanol for 3 days each to completely wash off impurities, if any. Then the product was freeze dried at −120 °C for 3 days.

**Material characterization**. Fourier transform infrared (FT-IR) spectra were obtained on a Spectrum 100 (Perkin-Elmer, USA) with a KBr pellet. Magic-angle spinning (MAS) nuclear magnetic resonance (NMR) spectra were measured at room temperature on an Agilent VNMRS 600 spectrometer. The thermogravimetric analysis (TGA) was carried out using a STA 8000 thermal analyzer at a heating rate of $10 \, °C \, min^{-1}$ under air and nitrogen atmosphere. Powder X-ray diffraction (PXRD) patterns were recorded using a High-Power X-Ray Diffractometer D/MAX 2500 V/PC (Cu-Kα radiation, 40 kV, 200 mA, $\lambda = 1.54056 \, Å$) (Rigaku Inc., Japan). Scanning electron microscope (SEM) measurements with Pt coated samples were obtained by a Field Emission Scanning Electron Microscope Nanonova 230 (FEI Inc., USA). X-ray photoelectron spectroscopy (XPS) was performed on an X-ray Photoelectron Spectrometer K-alpha (Thermo Fisher, UK). Elemental analysis (EA) data was obtained from a Flash 2000 Analyzer (Thermo Scientific Inc., USA). High-resolution transmission electron microscopy (HR-TEM) was conducted using a JEM-2100F microscope (JEOL inc., Japan) at an operating voltage of 200 keV. The samples were prepared by drop casting dispersed ethanol on a holey carbon TEM grid, and drying in an oven at 50 °C under vacuum.

**Low pressure gas adsorption study of samples (up to 1 bar)**. Low pressure gas adsorption measurements were carried out on samples which were heated at 150 °C for 12 h under dynamic vacuum to remove trapped solvents or moisture inside pores. Basic volumetric $N_2$ sorption studies were conducted at 77 K using the Brunauer-Emmett-Teller (BET) method on BELSORP-max (BEL Japan, Inc., Japan). Liquid nitrogen, liquid argon and ice-water baths were used to set temperatures of 77, 87, and 273 K, respectively. Ultra-high purity (UHP) grade $N_2$, Ar, and $CO_2$ gases (99.999% purity) were used for adsorption measurements. Oil-free environments (vacuum pump and pressure regulators) were used for all measurements to avoid contamination of the samples during the degassing process and isotherm calculations.

**High pressure gas adsorption study of the V2D-BBL structure**. The samples synthesized at UNIST were packed in an argon atmosphere glovebox before shipping to NIST for high pressure sorption measurement. Prior to the sorption study, the samples were activated as in the case of the low-pressure isotherms and the sample quality was checked using nitrogen isotherms. High pressure adsorption measurements were performed using our custom developed a fully computer-controlled Sieverts apparatus as discussed in detail in the literature[44]. Briefly, our fully computer-controlled Sievert apparatus operates in a sample temperature range of 20–500 K and a pressure range of 0 to 120 bar. In the volumetric method, gas with a known volume is admitted from a dosing cell to the sample cell in a controlled manner; the gas pressure and temperature are controlled and recorded.

Some unique features of our setup are as follows. We have five gas inlets including He, $N_2$, $CO_2$, $CH_4$, and $H_2$, enabling us to perform first nitrogen pore volume and surface measurements and then He-cold volume determination, and then the gas adsorption measurements, using the same protocol and without having to move the sample from the cell. We used two high precision digiquartz pressure gauges with parts-per-billion resolution and a typical accuracy of 0.01% (20 psia and 3000 psia, respectively) to precisely measure the pressure.

For isotherm measurements below room temperature, the sample temperature was controlled using a closed cycle refrigerator (CCR). The difference between the real sample temperature and the control set-point was within 1 K over the entire operating temperature range. The connection between the sample cell and the dose cell was through 1/8″ capillary high-pressure tubing, which provided a sharp temperature interface between the sample temperature and the dose temperature (i.e., room temperature). The cold volumes for the empty cell were determined using He as a function of pressure at every temperature before the real sample measurement and were used to calculate sample adsorption.

Since the adsorbed amount is deducted from the raw P-V-T data using a real gas equation of state, the accuracy of the chosen equation of state (EOS) is a critically important issue in terms of describing the real gas behavior within the desired temperature and pressure range. Using an empty cell as a reference, we found that the MBWR EOS best described the real gas behavior of He, $H_2$ and $CH_4$. Therefore, in all our isotherm data reduction, the NIST MBWR EOS was used. [NIST Standard Reference Database 23: NIST Reference Fluid Thermodynamic and Transport Properties Database].

**General procedures for iodine capture experiments**. Iodine vapor capture experiments were conducted using gravimetric methods at ambient pressure and 348 K to form an $I_2$@V2D-BBL structure complex. About 30 mg of the V2D-BBL structure was exposed to excess iodine vapor in a sealed chamber. The $I_2$@V2D-BBL structure complex samples were recovered after different exposure times and allowed to cool down to room temperature. To get more reliable capture amounts, the sample was heated to 333 K in a sealed chamber for 2 h to remove any iodine on the surface of the sample. Then, the sample weight was recorded after cooling down to room temperature.

The iodine sorption capacity was calculated by $(m_t - m_0)/m_0 \times 100 \, wt\%$, where $m_t$ is the mass of the $I_2$@V2D-BBL structure complex at certain time, and $m_0$ is the initial weight of the pristine V2D-BBL structure.

## Data availability

The data that support the findings of this study are available from the corresponding author upon reasonable request.

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

## Acknowledgements

This research was supported by the Creative Research Initiative (CRI, 2014R1A3A2069102), BK21 Plus (10Z20130011057), Science Research Center (SRC, 2016R1A5A1009405), and Young Researcher (2019R1C1C1006650) programs through the National Research Foundation (NRF) of Korea.

## Author contributions

J.B.B. and J.M. conceived and designed the project. H-J.N. and J.M. carried out the synthesis and characterization. Y-K.I., S-Y.Y., and J-M.S. performed microscopic characterizations. T.Y. carried out all the high-pressure gas adsorption studies. H-J.N., J.M. and J-B.B. wrote the paper and all authors discussed the results and commented on the paper.

## Competing interests

The authors declare no competing interests.
