## [Peer Review File · Nature Communications]

Reviewers' Comments:

Reviewer #1:

Remarks to the Author:

The manuscript entitled "Vertical two-dimensional layered fused aromatic ladder structure" (NCOMMS-20-03489-T) synthesized a 'vertical' 2D layered material with a robust fused aromatic ladder (FAL) structure. The vertical 2D layered FAL structure has excellent gas uptake performance under both low and high pressures, and also a high iodine (I₂) uptake capacity with unusually fast kinetics. The manuscript is well structured and the mechanisms of gas adsorption-desorption process are provided in detail. This study is of great importance and it promotes the development of the reclamation of gas uptake. Overall, the experiments have been conducted adequately. I believe that the paper is worth to be published in Nature Communications after minor revision based on the following:

1. - Page 3, "In particular, robust 2D fused aromatic network (FAN) structures, with rigid and fully π -conjugated linkages, are attractive for diverse applications^{9,13,14}". The author should elaborate on the research content and results of one of these mentioned examples.
2. Several expressions in the article need to be supported by related references. For example:
 - Page 3, "To date, however, reported 2D COFs, PONs and FANs still have mostly planar structures, with building block units which are flat, and lie parallel to the layer stacking direction, to optimize interlayer π - π stacking."
 - Page 6, "The appearance of the C=N stretching band at 1628 cm⁻¹ supports the formation of the benzimidazole ring in the BBL structure (Figure 2a)."
 - Page 8, "Both samples have three similar broad peaks with chemical shifts of 114 (c, d and c', d'), 128 (b, g and b', g') and 160 ppm (f and f'), which can be assigned to the aromatic sp² carbons and carbonyl (C=O) groups in their structure"
 - Page 8, "The up-field shift of the V2D-BBL structure, which suggests more deshielding, can be attributed to the inductive effect from the formation of fused aromatic rings and two electronegative nitrogen atoms in each benzimidazole ring"
 - Page 8, "Deconvolution of the N 1s resulted in two peaks, associated with the tertiary N and C=N-C at 400.6 and 398.5 eV, respectively." (Chem. Eng. J. 370 (2019) 1087-1100.)
3. More detailed information should be added at "In the present work, we report the synthesis of a 'vertical' 2D layered BBL (designated V2D-BBL) structure, and its unique sorption behaviors, which can be exploited as a new class of 2D structure." in Page 4.
4. - Page 9, "Elemental mapping showed that carbon, nitrogen and oxygen elements were well-distributed in the structure (Supplementary Figure 5)." It is recommended to use elemental chemical formula to express directly.
5. - Page 9, the sentence "Still, the local ordering with broad peaks, which is quite different from the interlayer interactions of planar 2D layered materials such as graphite and h-BN, suggests that the peak broadening may be, in significant vibration of structural units, which are vertically laid-up in the BBL structure and thus have a relatively high degree of freedom. part, associated with a randomly oriented stacking nature, and scattering induced by the" is too long.
6. - Page 15, "Moreover, the regenerated sample exhibited the same FT-IR spectrum as the pristine sample even after 5 times of reuse." The Figure should be provided.

Reviewer #2:

Remarks to the Author:

This manuscript reports the design and synthesis of vertically standing 2D BBL type of structure. Because of unique structural nature, this vertical 2D layered BBL framework presents high gas uptake efficiency in both low and high pressures. More importantly, it shows unusually high iodine uptake capacity and fast kinetics compared to other organic porous polymers. It seems the fast iodine uptake kinetics of is related to the vertically standing 2D structure, as explained by the authors. The presentation of the experimental results and characterizations of the materials have been well organized in a nice flow to support the discussion. This paper is certainly of great

general interest and significance in field of porous organic structures and framework syntheses. Therefore, I recommend it for publication in Nature Communications after minor revisions noted below:

1. In Figure 2b, ^{13}C -NMR assignment of 'e' to amine attached carbon is erroneous. The authors need to reassign the carbon peaks to correct structures.
2. It will be better to add SEM and TEM images of the material in the Supporting Information. As far as I know, it is hard to capture good TEM images because of vibration associated with lateral flexibility, but it will help the readers to understand properties of these materials.
3. Why V2D-BBI and PIN show comparatively different pore size distribution and quite different nitrogen isotherm as well, even though the base structure is almost the same.
4. The authors need to explain the difference in elemental composition values from different techniques (XPS, EDS) at the bottom of the Supplementary Table 1 for better understanding.
5. The yields of the polycondensation reaction between THA and NDA in polyphosphoric acid should be given. Interestingly, as the authors have prepared the samples in gram scale, that is quite important.
6. The authors have shown on I3d deconvoluted spectrum after iodine capturing, it will be good to show the survey XPS of I2@V2D-BBL as well.
7. Usually, it is hard to completely remove the phosphorus moieties after reaction. Did the authors check if there any P doping from the PPA?

Reviewers' comments:

Reviewer #1 (Remarks to the Author):

The manuscript entitled "Vertical two-dimensional layered fused aromatic ladder structure" (NCOMMS-20-03489-T) synthesized a 'vertical' 2D layered material with a robust fused aromatic ladder (FAL) structure. the vertical 2D layered FAL structure has excellent gas uptake performance under both low and high pressures, and also a high iodine (I₂) uptake capacity with unusually fast kinetics. The manuscript is well structured and the mechanisms of gas adsorption-desorption process are provided in detail. This study is of great importance and it promotes the development of the reclamation of gas uptake. Overall, the experiments have been conducted adequately. I believe that the paper is worth to be published in Nature Communications after minor revision based on the following:

Comment 1.1. - Page 3, "In particular, robust 2D fused aromatic network (FAN) structures, with rigid and fully π -conjugated linkages, are attractive for diverse applications^{9,13,14}". The author should elaborate on the research content and results of one of these mentioned examples.

Response 1.1. We are grateful to the reviewer #1 for his/her thoughtful evaluation of our manuscript and suggesting important changes to enhance the quality of the manuscript. Based on the reviewer's suggestions, we have added more detailed explanation of mentioned references in the revised manuscript.

Comment 1.2. Several expressions in the article need to be supported by related references. For example:

- Page 3, "To date, however, reported 2D COFs, PONs and FANs still have mostly planar structures, with building block units which are flat, and lie parallel to the layer stacking direction, to optimize interlayer π - π stacking."
- Page 6, "The appearance of the C=N stretching band at 1628 cm⁻¹ supports the formation of the benzimidazole ring in the BBL structure (Figure 2a)."
- Page 8, "Both samples have three similar broad peaks with chemical shifts of 114 (c, d and c', d'), 128 (b, g and b', g') and 160 ppm (f and f'), which can be assigned to the aromatic

sp² carbons and carbonyl (C=O) groups in their structure”

- Page 8, “The up-field shift of the V2D-BBL structure, which suggests more deshielding, can be attributed to the inductive effect from the formation of fused aromatic rings and two electronegative nitrogen atoms in each benzimidazole ring”

- Page 8, “Deconvolution of the N 1s resulted in two peaks, associated with the tertiary N and C=N-C at 400.6 and 398.5 eV, respectively.” (Chem. Eng. J. 370 (2019) 1087-1100.)

Response 1.2. Thanks for the pointing out important issues related to referencing. We have added related references in the revised manuscript to improve the literature survey. With due apology, we would like to mention that the type of nitrogen discussed in the given reference is quite different from our structure. However, we have added another more related reference that contains well-matched deconvoluted N 1s spectrum (XPS).

Comment 1.3. More detailed information should be added at “In the present work, we report the synthesis of a ‘vertical’ 2D layered BBL (designated V2D-BBL) structure, and its unique sorption behaviors, which can be exploited as a new class of 2D structure.” in Page 4.

Response 1.3. Based on the reviewer’s recommendation, we have added detailed elaboration of present work in the mentioned part of the revised manuscript (page 5).

Comment 1.4. - Page 9, “Elemental mapping showed that carbon, nitrogen and oxygen elements were well-distributed in the structure (Supplementary Figure 5).” It is recommended to use elemental chemical formula to express directly.

Response 1.4. Thanks to the reviewer for helpful suggestion to improve value of our work. As the reviewer recommended, EDS spectrum with chemical composition table is added in **Supplementary Figure 5 (Figure R1)**.

Figure R1. **a**, SEM image with EDS spectrum and elemental compositions of V2D-BBL structure. Scale bar: 2 μm. Corresponding elemental mappings: **b**, carbon, **c**, nitrogen and **d**, oxygen.

Comment 1.5. - Page 9, the sentence “Still, the local ordering with broad peaks, which is quite different from the interlayer interactions of planar 2D layered materials such as graphite and h-BN, suggests that the peak broadening may be, in significant vibration of structural units, which are vertically laid-up in the BBL structure and thus have a relatively high degree of freedom. part, associated with a randomly oriented stacking nature, and scattering induced by the” is too long.

Response 1.5. We agree with the reviewer. The length of sentence is too long. Hence, we have corrected this sentence in the revised manuscript.

Comment 1.6. - Page 15, “Moreover, the regenerated sample exhibited the same FT-IR spectrum as the pristine sample even after 5 times of reuse.” The Figure should be provided.

Response 1.6. The reviewer may have not noticed **Figure 4f**, in which we have given FT-IR spectra of the V2D-BBL structure (black line), I₂@V2D-BBL structure (purple line) and regenerated V2D-BBL structure after 5 recycles (grey line) in the range of 1300 to 1800 cm⁻¹. There is no noticeable changes of main peaks between V2D-BBL structure and regenerated V2D-BBL structure, indicating reversible nature of sample. For further confirmation of reversibility, the full FT-IR spectra of samples also given for the reviewer’s consideration below (**Figure R2**).

Figure R2. FT-IR spectra (KBr pellets) of the pristine V2D-BBL structure, I₂@V2D-BBL structure and regenerated V2D-BBL structure after 5 recycles.

Reviewer #2 (Remarks to the Author):

Comments:

This manuscript reports the design and synthesis of vertically standing 2D BBL type of structure. Because of unique structural nature, this vertical 2D layered BBL framework presents high gas uptake efficiency in both low and high pressures. More importantly, it shows unusually high iodine uptake capacity and fast kinetics compared to other organic porous polymers. It seems the fast iodine uptake kinetics of is related to the vertically standing 2D structure, as explained by the authors. The presentation of the experimental results and characterizations of the materials have been well organized in a nice flow to support the discussion. This paper is certainly of great general interest and significance in field of porous organic structures and framework syntheses. Therefore, I recommend it for publication in Nature Communications after minor revisions noted below:

Comments 2.1. In Figure 2b, ^{13}C -NMR assignment of ‘e’ to amine attached carbon is erroneous. The authors need to reassign the carbon peaks to correct structures.

Response 2.1. We appreciate the reviewer #2 for his/her thoughtful comments and critical assessment of our manuscript. The reviewer’s suggestions and criticisms help us to substantially improve the quality of manuscript. Based on the reviewer’s suggestion, we revised discussion for **Figure 2b** and highlighted in the revised manuscript.

Comments 2.2. It will be better to add SEM and TEM images of the material in the Supporting Information. As far as I know, it is hard to capture good TEM images because of vibration associated with lateral flexibility, but it will help the readers to understand properties of these materials.

Response 2.2. Thanks to the reviewer for important suggestion. As the reviewer mentioned, it was hard to get crystalline TEM image due to vibration corresponding to vertical 2D structure (more vibration e-beam). Nevertheless, we have examined the sheet-like texture with clean layer in SEM and HR-TEM. Both SEM and TEM representative images of the material in **Supplementary Figure 6**.

Comments 2.3. Why V2D-BBI and PIN show comparatively different pore size distribution and quite different nitrogen isotherm as well, even though the base structure is almost the same.

Response 2.3. We highly appreciate the reviewer's thoughtful question. PIN structure has a rotatable C-N single bond moiety, which makes the structure flexible. Thus, PIN structure exhibited relatively low hysteresis in N₂ isotherm with larger pore size from pore size distribution plot. Meanwhile, V2D-BBL structure is composed of fully fused aromatic system, which makes non-rotatable rigid backbone in the structure. Hence, narrow pores with large hysteresis were shown as inherent nature of V2D-BBL structure.

Comments 2.4. The authors need to explain the difference in elemental composition values from different techniques (XPS, EDS) at the bottom of the Supplementary Table 1 for better understanding.

Response 2.4. Once again, we appreciate the reviewer for valuable suggestion. As we have mentioned at the bottom of the **Supplementary Table 1**, XPS and SEM EDS are subjective techniques and mostly used for qualitative evaluation. Thus, both techniques are more sensitive to surface chemical composition, making the difference in elemental composition values. On the other hand, EA is more reliable technique for elemental counts for bulk sample and well-matched with expected values.

Comments 2.5. The yields of the polycondensation reaction between THA and NDA in polyphosphoric acid should be given. Interestingly, as the authors have prepared the samples in gram scale, that is quite important.

Response 2.5. Thanks to the reviewer for highlighting important point. Because of strong driving force for aromatization, the yields of these reactions are quantitative. In this particular case, when we carried out polycondensation between triptycene hexamine (1.0 g, 1.775 mmol) and 1,4,5,6-naphthalenetetracarboxylic dianhydride (0.714 g, 2.66 mmol) in PPA, the isolated yield was 1.35 g (99.1% after complete work-up procedures to remove any bound small molecules) for V2D-BBL structure.

Comments 2.6. The authors have shown on I3d deconvoluted spectrum after iodine capturing, it will be good to show the survey XPS of I2@V2D-BBL as well.

Response 2.6. For the reviewer's consideration, XPS survey spectrum of I₂@V2D-BBL structure is given below in **Figure R3**, which is added in **Supplementary Figure 19**.

Figure R3. XPS survey spectrum of I₂@V2D-BBL structure.

Comments 2.7. Usually, it is hard to completely remove the phosphorus moieties after reaction. Did the authors check if there any P doping from the PPA?

Response 2.7. We agree with the reviewer. Sometimes, it is hard to completely remove phosphorous (P). However, in this case, all possible P sources from PPA was removed by Soxhlet extraction with water and methanol for 3 days each. After complete work-up procedures, we confirmed that there is no P peak in XPS survey in **Figure 2c** and SEM-EDS spectrum (**Figure R4**).

Figure R4. SEM-energy dispersive (EDS) spectrum from SEM image of the V2D-BBL structure in **Supplementary Figure 5a**.

Reviewers' Comments:

Reviewer #1 :

Reviewer #1 made remarks to the editor only and supports publication of the manuscript after minor revision. The reviewer thinks that the study is important and that the experiments have been done adequately.

Reviewer #1 asks to:

* elaborate on one of the examples in the line In particular, robust 2D fused aromatic network (FAN) structures, with rigid and fully π -conjugated linkages, are attractive for diverse applications

* support claims in the manuscript by additional references, such as the claim that 2D COFs, PONs and FANs have mostly planar structures with flat building block units that lie parallel to the layer staking direction in order to optimize π - π stacking (pg. 3)

* or that the appearance of the C=N stretching band at 1620 cm^{-1} supports the formation of the benzimidazole ring in the BBL structure, as stated in Fig. 2a. on pg. 6.

* as well as on the claim that both samples have three similar broad peaks with chemical shifts of 114, 128 and 160 ppm which can be assigned to the aromatic sp^2 carbons and carbonyl (C=O) groups on pg. 8.

* as well as on "The up-field shift of the V2D-BBL structure, which suggests more deshielding, can be attributed to the inductive effect from the formation of fused aromatic rings and two electronegative nitrogen atoms in each benzimidazole ring" and "Deconvolution of the N 1s resulted in two peaks, associated with the tertiary N and C=N-C at 400.6 and 398.5 eV, respectively." (Chem. Eng. J. 370 (2019) 1087-1100.) on page 8.

* Please also provide some more information on the claim "In the present work, we report the synthesis of a 'vertical' 2D layered BBL (designated V2D-BBL) structure, and its unique sorption behaviors, which can be exploited as a new class of 2D structure." in Page 4.

* Furthermore, using the elemental chemical formula should be given for the elemental mapping on pg. 9 Suppl. Fig. 5 ("Elemental mapping showed that carbon, nitrogen and oxygen elements were well-distributed in the structure (Supplementary Figure 5).")

* Shortening the sentence "Still, the local ordering with broad peaks, which is quite different from the interlayer interactions of planar 2D layered materials such as graphite and h-BN, suggests that the peak broadening may be, in significant vibration of structural units, which are vertically laid-up in the BBL structure and thus have a relatively high degree of freedom. part, associated with a randomly oriented stacking nature, and scattering induced by the" on page 9 would facilitate the reading

* and finally, a figure should be provided on page 15 to support the statement "Moreover, the regenerated sample exhibited the same FT-IR spectrum as the pristine sample even after 5 times of reuse."

Reviewer #2:

Remarks to the Author:

In this revised version, the authors fully addressed my earlier comments and significantly improved the manuscript. Thus, i recommend publication as is.

Reviewers' comments:

Reviewer #1 (Remarks to the Author):

Reviewer #1 made remarks to the editor only and supports publication of the manuscript. The reviewer thinks that the study is important and that the experiments have been done adequately.

With respect to the comments of reviewer #1 to the editor I would like to ask you to:

Comment 1.1. Elaborate on one of the examples in the line In particular, robust 2D fused aromatic network (FAN) structures, with rigid and fully π -conjugated linkages, are attractive for diverse applications

Response 1.1. We are grateful to the reviewer for thoughtful evaluation of our manuscript and suggesting important changes to enhance the quality of the manuscript. Based on the reviewer's suggestion, we have added more detailed explanation of mentioned references into the revised manuscript.

Comment 1.2. Support claims in the manuscript by additional references, such as the claim that 2D COFs, PONs and FANs have mostly planar structures with flat building block units that lie parallel to the layer staking direction in order to optimize pi-pi stacking (pg. 3)

Response 1.2. We appreciate the reviewer for constructive suggestion. We have added additional suitable references to revised manuscript to support the text.

Comment 1.3. or that the appearance of the C=N stretching band at 1620 cm⁻¹ supports the formation of the benzimidazole ring in the BBL structure, as stated in Fig. 2a. on pg. 6.

Response 1.3. Thanks to the reviewer for the nice suggestion, we have added related references to revised manuscript.

Comment 1.4. as well as on the claim that both samples have three similar broad peaks with chemical shifts of 114, 128 and 160 ppm which can be assigned to the aromatic sp² carbons and carbonyl (C=O) groups on pg. 8.

Response 1.4. We are grateful to the reviewer for thoughtful suggestion. We have added

relevant references in the revised manuscript to enhance quality of the work.

Comment 1.5. as well as on “The up-field shift of the V2D-BBL structure, which suggests more deshielding, can be attributed to the inductive effect from the formation of fused aromatic rings and two electronegative nitrogen atoms in each benzimidazole ring” and “Deconvolution of the N 1s resulted in two peaks, associated with the tertiary N and C=N-C at 400.6 and 398.5 eV, respectively.” (Chem. Eng. J. 370 (2019) 1087-1100.) on page 8.

Response 1.5. Thanks for the pointing out important issues related to referencing, we have added related reference in the revised manuscript to improve the literature survey. With due apology, we would like to mention that the type of nitrogen discussed in the given reference is quite different from our structure. However, we have added another more pertinent reference which contains well-matched deconvoluted (XPS) N 1s spectrum.

Comment 1.6. Please also provide some more information on the claim “In the present work, we report the synthesis of a ‘vertical’ 2D layered BBL (designated V2D-BBL) structure, and its unique sorption behaviors, which can be exploited as a new class of 2D structure.” in Page 4.

Response 1.6. Based on the reviewer’s recommendation, we have added detailed elaboration of present work in the part of the revised manuscript.

Comment 1.7. Furthermore, using the elemental chemical formula should be given for the elemental mapping on pg. 9 Suppl. Fig. 5 (“Elemental mapping showed that carbon, nitrogen and oxygen elements were well-distributed in the structure (Supplementary Figure 5).”)

Response 1.7. Thanks to the reviewer for helpful suggestion to improve value of our work. As the reviewer recommended, EDS spectrum with chemical composition table is added in **Supplementary Figure 5 (Figure R1)**.

Figure R1. **a**, EDS spectrum with corresponding table revealing elemental compositions and SEM elemental mappings of V2D-BBL structure: **b**, carbon, **c**, nitrogen and **d**, oxygen.

Comment 1.8. Shortening the sentence “Still, the local ordering with broad peaks, which is quite different from the interlayer interactions of planar 2D layered materials such as graphite and h-BN, suggests that the peak broadening may be, in significant vibration of structural units, which are vertically laid-up in the BBL structure and thus have a relatively high degree of freedom. part, associated with a randomly oriented stacking nature, and scattering induced by the” on page 9 would facilitate the reading.

Response 1.8. We agree with the reviewer, the length of sentence is too long. Hence, we have corrected this sentence in the revised manuscript.

Comment 1.9. And finally, a figure should be provided on page 15 to support the statement “Moreover, the regenerated sample exhibited the same FT-IR spectrum as the pristine sample even after 5 times of reuse.

Response 1.9. The reviewer may have not noticed that we have given FT-IR spectra of the V2D-BBL structure (black line) in **Figure 4f**: I₂@V2D-BBL structure (purple line) and

regenerated V2D-BBL structure after 5 recycles (grey line) in the range of 1300 to 1800 cm^{-1} . Both V2D-BBL structure and regenerated V2D-BBL structure display unchanged main peaks, indicating reversible nature of sample. For further confirmation of reversibility, the full FT-IR spectra of samples also given for the reviewer's consideration (**Figure R2**).

Figure R2. FT-IR spectra (KBr pellets) of the pristine V2D-BBL structure, I₂@V2D-BBL structure and regenerated V2D-BBL structure after 5 recycles.

Reviewer #2 (Remarks to the Author):

Comments:

In this revised version, the authors fully addressed my earlier comments and significantly improved the manuscript. Thus, i recommend publication as is.

Response. We appreciate the reviewer #2 for recognizing the importance of our work and thoughtful comments with critical assessment of our manuscript. The reviewer’s suggestions and criticisms help us to substantially improve the quality of manuscript.